# A Twist in Yeast: New Perspectives for Studying TDP-43 Proteinopathies in *S. cerevisiae*

**DOI:** 10.3390/jof11030188

**Published:** 2025-02-28

**Authors:** Roberto Stella, Alessandro Bertoli, Raffaele Lopreiato, Caterina Peggion

**Affiliations:** 1Laboratorio Farmaci Veterinari e Ricerca, Istituto Zooprofilattico Sperimentale delle Venezie, 35020 Legnaro, Italy; rstella@izsvenezie.it; 2Department of Biomedical Sciences, University of Padova, 35131 Padova, Italy; alessandro.bertoli@unipd.it (A.B.); raffaele.lopreiato@unipd.it (R.L.); 3Neuroscience Institute, Consiglio Nazionale Delle Ricerche, 35131 Padova, Italy; 4Padova Neuroscience Center, University of Padova, 35131 Padova, Italy; 5Department of Biology, University of Padova, 35131 Padova, Italy

**Keywords:** TDP-43, neurodegeneration, RNA metabolism, protein aggregation, ALS, chaperone, yeast, autophagy, mitochondrial dysfunction, nucleolin

## Abstract

TAR DNA-binding protein 43 kDa (TDP-43) proteinopathies are a group of neurodegenerative diseases (NDs) characterized by the abnormal accumulation of the TDP-43 protein in neurons and glial cells. These proteinopathies are associated with several NDs, including amyotrophic lateral sclerosis, frontotemporal lobar degeneration, and some forms of Alzheimer’s disease. Yeast models have proven valuable in ND research due to their simplicity, genetic tractability, and the conservation of many cellular processes shared with higher eukaryotes. For several decades, *Saccharomyces cerevisiae* has been used as a model organism to study the behavior and toxicity of TDP-43, facilitating the identification of genes and pathways that either exacerbate or mitigate its toxic effects. This review will discuss evidence showing that yeast models of TDP-43 exhibit defects in proteostasis, mitochondrial function, autophagy, and RNA metabolism, which are key features of TDP-43-related NDs. Additionally, we will explore how modulating proteins involved in these processes reduce TDP-43 toxicity, aiding in restoring normal TDP-43 function or preventing its pathological aggregation. These findings highlight potential therapeutic targets for the treatment of TDP-43-related diseases.

## 1. *S. cerevisiae*: A Model for Studying Neurodegenerative Diseases

The yeast *Saccharomyces cerevisiae* has made significant contributions to our understanding of fundamental and well-conserved eukaryotic biology, including cell cycle regulation and autophagy [1,2,3]. Interestingly, many aspects of evolutionarily conserved mechanisms of mitochondrial dynamics, and mitochondrial ion transporter, are derived from research on *S. cerevisiae* [4]. There are many reasons for this, primarily the high degree of genetic conservation, the smaller genome, and fewer gene duplications in yeast cells compared to higher eukaryotes, facilitating the study of biological and biochemical functions occurring in more evolved organisms.

Additionally, its simplicity in performing genetic mutations/deletions in both haploid and diploid forms, its rapid growth, low cost of introduction and maintenance, and the absence of significant ethical concerns made it an attractive model organism. Another advantage is that it was the first eukaryotic organism to have its entire genome sequenced [5,6], enabling scientists to develop new genomic techniques to tackle a wide range of challenges. Furthermore, it is worth mentioning that the expansion of genetic and molecular techniques (with the construction of a comprehensive collection of yeast deletion mutants [7], genome-wide overexpression libraries, and green fluorescent protein-tagged yeast strains [8]), the development of high-throughput technologies to study the transcriptome [9], the proteome [10], and the metabolome [11,12], made yeast a popular organism for “systems modeling”. Furthermore, yeast is highly valuable for drug discovery and testing due to its suitability for high-throughput screening. It is also useful for identifying the mechanism of action of compounds with unknown targets, and it represents a unique system where all cellular targets can be simultaneously assessed in vivo [13,14,15,16]. Although yeast is highly valuable for many aspects of research, it also presents limitations for investigating intricate biological processes due to the following reasons: (i) it lacks the complexity of multicellular organisms and tissue-specific regulation, which limits its ability to fully replicate complex biological systems; (ii) its simpler or divergent signaling pathways make it more challenging to study certain disease mechanisms; (iii) it has a limited capacity for post-translational modifications, which are crucial for regulating complex biological processes; and (iv) its metabolism is simple and cannot totally reproduce mammalian models. In summary, while yeast is a powerful tool for high-throughput screening and studying simpler biological mechanisms, it may not capture the full intricacies of complex processes, particularly those involving multi-layered regulation, tissue-specific factors, or advanced post-translational modifications [17].

However, with its strength and weaknesses, *S. cerevisiae* has a long-standing and successful history in modeling a wide range of human diseases as well as in unraveling disease-associated human genetic variants of genes encoding proteins involved in various pathways. The study of yeast, combined with the analysis of patient-derived cells, has proven crucial in understanding the significance of these genetic variants.

The basic question is how can human diseases be studied in yeast? There are two approaches to achieve this goal. The first applies to cases where the disease-associated gene has an orthologue in yeast cells. In this case, information about the gene can be found by searching the Yeast Genome Database (SGD Project; http://www.yeastgenome.org/ (accessed on 10 January 2025)). If no information is available, the most suitable strategies are either gene deletion or its replacement with the mutated isoform (today the CRISPR/Cas9 technique makes these processes quick and efficient [18]). However, if the gene responsible for the disease does not have a corresponding orthologue in yeast, a model can still be created. This can be achieved by heterologously expressing the human gene (either mutated or not) in yeast cells, resulting in ’humanized’ strains. Importantly, both approaches have been successfully applied to define the phenotype resulting from the expression of human disease-associated genes in yeast or to perform functional studies of yeast orthologues of human disease-related genes [19] (Figure 1).

Over the last three decades, *S. cerevisiae* has emerged also as an effective model for studying neurodegenerative disorders (NDs), including Huntington’s disease, Parkinson’s disease (PD), amyotrophic lateral sclerosis (ALS), and Alzheimer’s disease (AD) and a variety of highly feasible yeast models have been created [20,21,22]. The first was specifically established in 2000 by Susan Lindquist’s group, which later became a valuable model for studying amyloid-associated disorders and the hallmarks of amyloid formation and propagation [23]. Also, in their pioneering study, Krobitsch and Lindquist established a model system to investigate neurodegeneration caused by polyglutamine (polyQ) expansions in the huntingtin (Htt) protein. They engineered yeast cells to express Htt with varying polyQ repeat lengths, demonstrating that the extent of aggregation correlated with the length of the polyQ repeat, mimicking the pathology observed in human neurons. Lindquist’s team also discovered that Hsp104, a yeast chaperone protein (see Section 2.4), could disassemble these aggregates, highlighting the potential for chaperone-based therapies in treating neurodegenerative diseases [24]. NDs are characterized by the progressive loss of neurons in different brain areas, often associated with the misfolding and aggregation of disease-specific proteins. Although yeast cells lack neuron-specific structures and functions, many basic metabolic pathways involved in neurodegeneration are well conserved. This makes them a useful model for studying NDs, especially when examining processes such as the aggregation of abnormally folded proteins or the disruption of normal protein homeostasis. These alterations often lead to a loss of protein function or the acquisition of a toxic function, both of which are characteristic of NDs. Such a high degree of conservation makes it possible to model basic molecular mechanisms of NDs (see review [25]), allowing the identification of modifiers of protein aggregation and toxicity, often confirmed in other in vitro and in vivo models. This might help in the identification of new therapeutic targets and treatments.

Among NDs successfully studied in yeast, we will specifically focus this review on neurodegenerative mechanisms associated with TAR DNA-binding protein 43 kDA (TDP-43) proteinopathies, in particular ALS, emphasizing how the combination of research in yeast and higher organisms can contribute to the development of novel therapeutic strategies.

## 2. ALS and TDP-43

### 2.1. ALS Pathology and TDP-43

ALS is a progressive and fatal ND characterized by the loss of motor neurons (MNs) in the motor cortex, the brain stem nuclei, and the anterior horn of the spinal cord leading to muscular weakness and wasting [26,27]. In most patients, the disease is relentlessly progressive, with a median survival of approximately three-five years from symptom onset, and death is primarily due to respiratory failure [28]. While most ALS cases are classified as sporadic (sALS) due to the absence of a direct association with inherited mutations, approximately 10% of cases, known as familial ALS (fALS), are inherited in an autosomal dominant manner. Up to 17 genes have been implicated in the pathogenesis of fALS, and their mutations were linked to disease onset and increased susceptibility to ALS. The most significant of these are C9orf72, the Cu–Zn superoxide dismutase 1 enzyme (SOD1), TDP-43, and fused in sarcoma/translocated in sarcoma (FUS/TLS) [29,30,31]. Different cellular mechanisms have been implicated in MNs cell death in ALS, such as oxidative stress, endoplasmic reticulum stress, glutamate excitotoxicity, neuroinflammation, dysfunctional RNA metabolism and deposition, and accumulation of misfolded protein. As for the last issue, strong evidence suggests that the onset and progression of ALS pathology is strictly associated with the deposition of proteins into insoluble inclusions due to protein misfolding and aggregation [32,33,34,35,36,37,38,39]. In a majority of ALS cases, such inclusions are associated with TDP-43 positive inclusions, where the remainder are associated with either SOD1 (~2%) or FUS (~1%) inclusions [40]. Key evidence suggests a cell-to-cell propagation of such inclusions, thus proposing the spread of ALS disease pathology both via prion-like seeding of protein aggregation, as occurs for many other neurodegenerative disorders, as well as by inducing cellular stress and thereby propagating a “disease state” [41]. The fact that major ALS mutations have been identified in some important RNA-processing genes, along with the presence of abnormal aggregates of the two RNA-binding proteins TDP-43 and FUS/TLS in ALS-affected tissues, suggests a crucial role for RNA dysmetabolism in ALS pathogenesis. It remains unclear which of the two processes occurs first. However, strong evidence demonstrated the early perturbation of molecular mechanisms regulating RNA metabolism, such as the alteration of gene transcription, disruption of the global splicing machinery (in the case of TDP-43 and FUS/TLS), changes in the splicing pattern of specific RNAs (in the case of C9orf72), alterations in mRNA transport, and local translation and modifications in the production of microRNAs [42]. It must be considered that RNA dysmetabolism and altered proteostasis are just two of many cellular processes involved in ALS, making it difficult to determine which is causal.

Numerous ALS genetic animal models were generated in the past decades (see [43] for a review), and their use provided critical insights into disease pathophysiology. Inherent limitations of animal models must nevertheless be considered, particularly relating to the difficulty to analyze genetic and molecular issues, and the difficulty that mouse models often faced with the translation to human medicine [44,45,46]. In this context, *S. cerevisiae* is an ideal model to investigate the molecular mechanisms of cytotoxicity induced by ALS-related proteins. Over the past few decades, up to twenty yeast models expressing different ALS-associated proteins have been generated (see the review [47]). Most of these models involve the overexpression of genes linked to RNA metabolism and protein misfolding, such as TDP-43, FUS, and SOD1, among others.

### 2.2. TDP-43: Structure, Yeast Models of TDP-43 Proteinopathies, and the Identification of Therapeutic Modifiers in ALS

TDP-43, discovered in 1995 as an HIV-1 gene expression regulator [48], is a highly conserved member of the heteronuclear ribonucleotide-binding protein (hnRNP) family [49]. TDP-43, like all hnRNPs, binds to nascent pre-mRNA molecules as soon as they emerge from RNA polymerase II and regulates their processing steps either sequentially or in collaboration/antagonism with neighboring RNA-binding factors. Besides mRNA processing, TDP-43 has been shown to have a role also in the maturation of other RNA molecules, as its implication in the generation of non-coding RNAs (miRNAs, lncRNAs, etc.), whose function is still mostly unknown [50,51,52]. The gene coding for TDP-43 was found on human chromosome 1p36.22 and consists of six exons. Several TDP-43 isoforms have been identified, although they have not been thoroughly explored [53]. Similarly to other hnRNPs, TDP-43 structure consists of multiple independent domains. At the N-terminus, a well-folded domain is present, consisting of one helix and six β-sheets (NTD residues 1–103, Figure 2). NTD is necessary for TDP-43 dimerization, physiological interaction with other proteins, and to form dimers and higher-order reversible oligomers in a concentration-dependent manner [54,55,56]. These oligomers, which are distinct from hyperphosphorylated and pathological aggregates, are essential for TDP-43 splicing activity and for the physiological formation of cytoplasmic stress granules through the liquid-liquid phase separation (LLPS) process (see below) and liquid droplet formation [54]. Proximal to the NTD, a canonical positively charged bi-dentate nuclear localization signal (NLS) is present, which is recognized by importin-α for the active transport of TDP-43 into the nucleus [57]. NLS disruption by caspase 3 cleavage at Asp 89 or by initiation at the alternative start site Met 85 results in the accumulation of TDP-43 in the cytoplasm and in the formation of insoluble aggregates [58]. Residues 106–262 of TDP-43 contain two RNA Recognition Motif (RRM) domains, which are necessary for binding to UG/TG-rich single-stranded or double-stranded DNA/RNA. These domains play a role in transcriptional suppression, pre-mRNA alternative splicing, and translation control. Additionally, a putative nuclear export sequence (NES) was identified within the second of the two RRMs [52,59,60,61,62]. The C-terminal tail of TDP-43 contains a low complexity domain (LCD), which is critical for imparting prion-like characteristics to the protein. For example, the LCD sequence, enriched in glycine, asparagine, and glutamine, is a common feature found in many proteins, including RNA-binding proteins (RBPs) and is essential for TDP-43 to undergo LLPS. LLPS is a dynamic process in which biomolecules, such as proteins and RNAs, separate from a homogeneous solution into distinct membraneless, spherical compartments. These compartments, also known as ribonucleoprotein (RNP) granules, are essential for regulating various cellular functions, including RNA metabolism, stress responses, and protein synthesis. The ability of TDP-43 to undergo LLPS is pivotal in maintaining its solubility, folding, and functionality within the cell. Under normal physiological conditions, TDP-43 reversibly assembles into RNP granules that specifically incorporate RNA and other RBPs. These granules are thought to be involved in RNA processing, storage, and transport. However, when the balance between LLPS and aggregation is disturbed, TDP-43 can form pathological protein aggregates [56,63]. LLPS is not a static phenomenon; it is finely regulated and can be influenced by post-translational modifications and mutations, crucial in modulating the interaction between TDP-43 and other molecules. These modifications can affect the electrostatic interactions between the protein’s domains, altering its phase separation behavior. As a consequence, mutations in the LCD or other regions of TDP-43 can disrupt these electrostatic interactions, leading to the loss of phase separation and the promotion of abnormal aggregation, which is a central feature in the pathogenesis of ALS and other TDP-43 proteinopathies [64,65]. Consistently, although TDP-43 mutations account for only a small number of fALS cases, most of them (in total, 69 variants (https://alsod.ac.uk/output/gene.php/TARDBP, accessed on 30 January 2025) are located within the LCD and are associated with an increased aggregation propensity and protein half-life, altered sub-cellular localization, and protein–protein interactions [66,67].

In physiological conditions, TDP-43 resides in the nucleus, although it can shuttle to the cytoplasm to carry out all its diverse functions [50]. Its abundance and localization were reported to be finely regulated by a negative feedback mechanism [68], while acute stress induces TDP-43 to concentrate into the cytoplasm with other RNPs and RNAs forming cytoplasmic stress granules that could disassemble when stress conditions end [69]. On the other hand, continuous stress leads to pathological condition, and occurrence of pathogenic mutations may drive to the accumulation and aggregation of TDP-43 into cytoplasmic inclusions.

Despite the presence of TDP-43 inclusions in almost all ALS cases, implying that protein mislocalization and aggregation are important events for disease progression [35,70], it is unclear whether TDP-43-induced toxicity is caused by cytoplasmic toxic gain-of-function or nuclear loss-of-function [71,72,73,74]. It is worth mentioning that such inclusions are present not only in ALS but also in other NDs, including frontotemporal dementia and AD, overall being classified as TDP-43 proteinopathies [75].

**Figure 2 jof-11-00188-f002:**
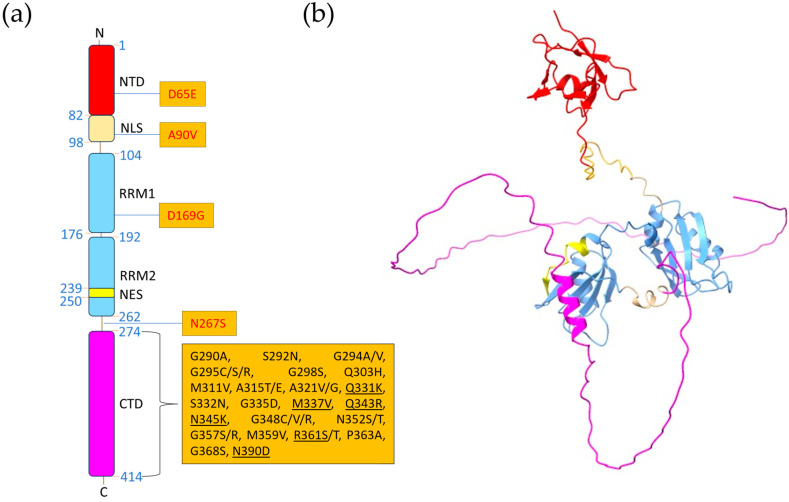
Human TDP-43 structure. (**a**) Illustration of TDP-43 primary structure, representing the different protein domains by colored rectangles. ALS-related mutations are indicated in the orange boxes. Mutations that were shown to increase the toxicity of TDP-43 in yeast models are reported in black. Underlined mutations are those confirmed to increase TDP-43 toxicity in yeast by different laboratories (see Section 2.3). (**b**) Model of the 3D structure of TDP-43 obtained with AlphaFold [76]. The model confidence of the N-terminal domain (NTD) (in red) and (RNA Recognition Motifs) RRMs domains (in light blue) is high. The nuclear localization signal (NLS) and the nuclear export sequence (NES) are represented in pink and yellow, respectively. The representation of the CTD (in purple) has a low level of confidence as this domain is disordered. The different domains of TDP-43 are represented in the same color used in panel (**a**).

### 2.3. Yeast Models of TDP-43 Proteinopathies

Several yeast models of TDP-43-related neurodegeneration were generated in the past decades, and thanks to the fact that yeast lacks bona fide TDP-43 orthologues, it had been possible to easily model TDP-43 toxicity by its ectopic overexpression in such a simple model organism. The first TDP-43 yeast model, generated by Gitler’s group, by the heterologous expression of human TDP-43 driven by a galactose-inducible promoter, provided precious information on mechanisms governing TDP-43 subcellular localization and aggregation [77]. Importantly, the yeast model recapitulated the most relevant pathological TDP-43 features, such as the cellular toxicity induced by the protein expression, as well as the TDP-43 cytoplasmic mislocalization and aggregation. Additionally, by expressing truncated forms of TDP-43, Gitler’s group identified specific regions within the protein that are both necessary and sufficient for nuclear localization, aggregation, and/or toxicity. Notably, they demonstrated that the CTD alone is insufficient for aggregation and toxicity; instead, an intact RRM2 is required for robust cytoplasmic foci formation and the induction of cell death [77]. Thereafter, such model has been successfully adopted to identify potent modifiers of TDP-43 toxicity using either genetic or chemical high-throughput screenings [78,79,80]. A pioneering effort was also another work of Gitler’s group, which investigated the effects of ALS-associated TDP-43 mutations in the CTD on protein aggregation and toxicity using the aforementioned TDP-43 yeast model. Their findings revealed that ALS-related mutations increase TDP-43 aggregation and promote toxicity in vivo. Additionally, mutations that were found to enhance toxicity in vivo also accelerate the aggregation of pure TDP-43 in vitro [64]. In contrast, Lehner’s group reported an opposing result, showing that mutations increasing hydrophobicity and aggregation tend to reduce toxicity, consistent with models suggesting that mature protein aggregates are not inherently toxic [81]. This conclusion was based on the overexpression in yeast cells of over 50,000 different TDP-43 mutants, including those found in both fALS and sALS patients. Using this approach, they examined the behavior of TDP-43 protein variants, including their propensity for LLPS and its effects on protein solubility and toxicity. Despite these findings, Bolognesi et al. also confirmed that, consistent with earlier studies, the expression of ALS-associated TDP-43 mutants remains more toxic in yeast cells than the wild-type protein, further reinforcing previous data [81]. ALS-related mutations (all of them in the CTD) that have been shown to increase TDP-43 toxicity in yeast by either Bolognesi et al. ([81], in black) or Gitler’s group ([77], underlined) are reported in Figure 2.

Other models of TDP-43 proteinopathy were generated in yeast, where the protein expression was transcriptionally regulated by different promoters, either constitutive or conditional, such as the copper-inducible promoter (CUP1), to precisely determine the relationship between TDP-43 protein amount and the extent of cellular damage [82]. Very recently, the CRISPR/Cas9 technique allowed generating fine-tuned yeast models by inserting into the cellular genome the human TDP-43 transgene, under the control of galactose-inducible promoter [83]. Interestingly, data revealed that the TDP-43 levels expressed by one chromosomal copy of the transgene did not affect cell survival, but the presence of two transgene copies led to the complete lethality of yeast strains. Notably, cytotoxicity and TDP-43 misfolding appeared directly correlated, as in severely damaged cells, the TDP-43 insoluble form was found at higher levels, along with the formation of TDP-43 inclusions [81]. These findings pointed to the crucial relevance of TDP-43 solubility to prevent protein misfolding and aggregation, which finally lead to cellular damage [83]. In the following sections, we will explore mechanisms discovered in yeast models that have been shown to reduce TDP-43 toxicity, including detoxifying processes such as the degradation of misfolded proteins, the rescue of misfolded proteins through chaperones, and the clearance of molecular aggregates via autophagy.

### 2.4. Chaperone Proteins as Modulators of TDP-43 Toxicity

In the last decade, several studies have reported that the toxicity associated with TDP-43 could be mitigated by chaperone proteins both in yeast and mammalian cells [84,85]. One such chaperone is Sis1 (Hsp40), the homolog of mammalian DNAJB1. When overexpressed in yeast cells, both Sis1 and DNAJB1 chaperones have been shown to restore ubiquitin-dependent proteolysis and alleviate the toxicity induced by TDP-43 overexpression and accumulation [86]. This result was further corroborated in cortical neurons, where DNAJB1 overexpression suppressed cell death induced by TDP-43 overexpression. Interestingly, this chaperone was also shown to mitigate the toxicity induced by the overexpression of FUS, which in turn inhibited the ubiquitin-proteasome system, similar to TDP-43 [86]. Probably DNAJB1 participates in maintaining TDP-43 in a soluble state. Additionally, both in yeast and mammalian neuronal cells, impaired Hsp90 function has been shown to worsen TDP-43 misfolding and sensitize cells to TDP-43 toxicity, while Sti1, a co-chaperone of Hsp90, by specifically interacting with TDP-43, was shown to modulate its toxicity in a dose-dependent manner [87]. Another essential chaperone that could be effective for rescue from toxicity induced by neurodegenerative proteins is the Hsp104 protein, a member of AAA+ protein disaggregases, which is highly conserved only in eubacteria and eukaryotes, though absent in metazoans [88,89,90,91]. The Hsp104 chaperone may promote the solubilization of various amyloids and prefibrillar oligomers, as well as enhance protein recovery from disordered aggregates after environmental stress [92,93,94,95], although its activity appeared to be limited against protein misfolding events that underlie neurodegenerative diseases [94]. James Shorter’s group created different Hsp104 variants using site-specific mutagenesis, which improved the protein’s ability to target disease-related proteins like TDP-43, FUS, and alpha-synuclein, while reducing unwanted side effects [96]. This enabled the creation of synthetic, non-toxic disaggregases, enhancing substrate specificity and tailoring specific therapeutic strategies for NDs [94,95]. Subsequently, the same group found that five Hsp104 homologs from two protozoa, two chromists, and the plant *Populus euphratica* could also reduce TDP-43 condensation and toxicity in yeast, as well as TDP-43 aggregation in Hek293T human cells. These natural Hsp104 homologs that reduce TDP-43 toxicity have little effect on alpha-synuclein and FUS toxicity, suggesting they work in a substrate-specific way [93]. These findings suggest that Hsp104, and potentially other AAA+ proteins, are uniquely equipped to counteract harmful protein misfolding and aggregation, offering promising therapeutic potential.

### 2.5. Autophagy and Its Impact on TDP-43 Toxicity

Autophagy is thought to play a critical role in clearing aggregated proteins, including TDP-43, from cells. Consequently, defective autophagy has been linked to the pathophysiology of several NDs [97]. In different cell models, TDP-43 down-regulation was shown to cause an impairment in the progression of the entire autophagic flux by modulating protein expression at multiple levels, from the transcription of autophagy-involved genes to the splicing and stabilization of mRNA molecules (see the review [97]). Consistently, increasing autophagy reduces TDP-43 toxicity both in primary neurons and in ALS mouse models [98,99]. Yeast models were explored to investigate how autophagy may influence TDP-43 aggregation and toxicity, and/or to identify potential therapeutic strategies to improve protein clearance through autophagy. Interestingly, the absence of two autophagy-related proteins, Pab1-binding protein 1 (Pbp1) (orthologue of the human ALS risk-related gene coding for Ataxin-2 [100]) and Tip41, has been demonstrated to alleviate TDP-43 toxicity by counteracting its inhibition of autophagy [101,102]. Notably, experimental evidence further indicated that the autophagy impairment due to TDP-43 expression in yeast cells could reside in the decreased formation of the autophagy enhancer TOROID (TORC1-Organized-in-Inhibited-Domain), although it has still to be confirmed in higher eukaryotes [103]. Such deleterious effect was shown to be mediated by the liquid-like but not amyloid-like TDP-43 aggregates in yeast, thus offering broad implications for understanding the involvement of protein misfolding processes in TDP-43-related neurodegeneration [104].

### 2.6. Mitochondrial Dysfunction and TDP-43 Toxicity

Since its facultative aerobic metabolism, the yeast *S. cerevisiae* may be extremely helpful in investigating mitochondrial function under both normal and stressful conditions. The ability to survive without functional mitochondria, as seen in ’petite’ mutants lacking mitochondrial DNA, makes yeast unique among other model organisms. So far, a series of non-fermentable carbon sources (e.g., glycerol, lactate) have been used to perform simple viability assays and determine the mitochondrial dysfunctions possibly caused by the expression of pathological proteins in yeast cells.

In ALS research, several studies have demonstrated that one of the affected pathways is mitochondrial functionality, in which TDP-43 may be a major player [105]. Notably, pathogenic TDP-43 was associated with increased mitochondrial localization and mitochondrial damage via the activation of the mitochondrial unfolded protein response pathway [106]. Recently, an integrative approach encompassing analysis of transcriptomic datasets from ALS patients and transgenic TDP-43, SOD1, and FUS ALS mouse models demonstrated that oxidative phosphorylation was a major deregulated pathway in the disease, with an enrichment of proteins and inhibitors of mitochondrial Complex III and IV [107]. Analyses in an ALS-related yeast model confirmed that these two complexes play a critical role in amyloidogenesis, suggesting them as potential therapeutic targets for ALS. Further studies have shown that fatty acid degradation, TCA cycle, and nicotinamide metabolism may also influence amyloidogenesis in TDP-43-related ALS [108]. Additionally, the impairment of mitochondrial respiratory capacity as a TDP-43 toxicity mechanism was confirmed in yeast, by observing a clear correlation between yeast respiration rate and TDP-43-induced cell death. Moreover, treatment with reactive oxygen species associated with respiration confirmed that the mitochondrial pathway increases TDP-43 toxicity [109,110]. More recently, researchers demonstrated that TDP-43 expression in yeast induced the translocation of Cnc1, homologous to human Cyclin C, to the cytoplasm, triggering a mitochondrial fragmentation-dependent cell death pathway, which is inhibited by antioxidants [111]. Interestingly, this rescue effect was observed only in the presence of functional mitochondria. Overall, these findings suggest that targeting the oxidative stress induced by TDP-43 expression and the translocation of Cyclin C may offer promising therapeutic avenues for ALS treatment [111].

### 2.7. RBPs as Modifiers of TDP-43 Toxicity

As cited above, RNA metabolism and dysregulation of RBPs activity are two principal disrupted processes in ALS and represent a promising therapeutic target. Besides autophagy (see Section 2.5), Ataxin-2 is now recognized to play multiple functions, ranging from the regulation of RNA stability and translation to the repression of deleterious accumulation of the RNA–DNA hybrid-harboring R-loop structures, also including stress granules dynamics, endocytosis, calcium signaling, and the regulation of the circadian rhythm [112]. Interestingly, it was demonstrated that TDP-43 physically and transiently interacts with Ataxin-2, with this association depending on the ability of TDP-43 to interact with RNA. Other works demonstrated that under stress conditions Ataxin-2 binds the 3′UTR of TDP-43 mRNA, leading to an increase in the transcript stability and protein levels [113]. Although the mechanism is still debated, Ataxin-2 knocking down prevents toxicity mediated by TDP-43 overexpression in several model systems [113], thus encouraging its modulation to prevent neurodegeneration.

Other two RNA-related proteins cure TDP-43 toxicity in yeast: the polyA-binding protein (Pabp) and the RNA lariat debranching enzyme (Dbr1) [78,114,115]. The first one was shown to interact with the previously discussed Ataxin-2 and modulate stress granule formation, while the second protein participates in intron splicing. In all such cases, it is plausible to hypothesize that by acting on RNA metabolism processes they could join into events triggering neurodegeneration.

We recently reported that the RNA/DNA-binding protein nucleolin (Ncl) can act as a potent suppressor of TDP-43 toxicity in yeast and mammalian cell models, by reducing the formation of detergent-insoluble assemblies and by avoiding the extra-nuclear accumulation of TDP-43 [83,116]. Interestingly, TDP-43 cytotoxicity was not alleviated by the overexpression of other human nucleolar proteins, such as Ncl functional partner nucleophosmin, or the yeast Ncl putative ortholog Nsr1p [83]. Although Ncl is primarily located in the dense fibrillar and granular components of the nucleolus, it can also shuttle between the nucleoplasm, cytoplasm, and plasma membranes, thereby taking part in a variety of cellular functions. These include the regulation of nucleolar assembly and function, chromatin remodeling, ribosomal RNA transcription and maturation, ribosome biogenesis and assembly, nucleocytoplasmic transport, and mRNA maturation. Additionally, Ncl exhibits helicase and self-cleaving activity and acts as an extracellular signal mediator at the cell surface (reviewed in [117]). A correlation between neurodegeneration and Ncl expression has been already established. In particular, Ncl expression was found to be reduced in patients affected by PD, while its overexpression enhances the clearance of aggregated PD-related alpha-synuclein in mouse embryonic fibroblasts and rat primary cortical neurons [118]. Additionally, the expansion of C9orf72 in ALS has been shown to disrupt Ncl’s nucleolar localization, exacerbating nucleolar stress and neuronal death [119]. Taken together, such findings strongly indicate Ncl as a promising therapeutic target. Modulating its activity could offer new therapeutic opportunities, either by enhancing its protective functions (such as its antioxidant properties) or by preventing its contribution to neurodegenerative pathways, including misfolded protein accumulation and neuroinflammation.

Overall, the use of yeast models of TDP-43 proteinopathy allowed to identify RBPs as suppressors of toxicity, but it could further provide relevant information concerning the molecular mechanisms of such suppressive effect, which are still to be determined, in yeast and mammalian cells.

### 2.8. TDP-43 Toxicity Modifiers Beyond RNA-Binding Proteins

In addition to the previously mentioned suppressors of TDP-43 toxicity, several others involved in various cellular functions have been identified through unbiased genome-wide yeast deletion screens ([78], as previously cited). For instance, genetic suppressors of TDP-43 toxicity include Cce1, a mitochondrial endonuclease involved in mitochondrial genome maintenance, and Dom34, a protein that facilitates the dissociation of inactive ribosomal subunits to enable translation re-initiation, particularly of ribosomes stalled at 3′ UTRs. While the human orthologue of Cce1 remains unknown, Dom34′s human counterpart is PELO (pelota mRNA surveillance and ribosome rescue factor). However, no evidence yet supports the role of PELO in rescuing TDP-43 toxicity in humans. Additionally, the deletion of *RPL16A*, a component of the large 60S ribosomal subunit, or *SIW4*, a gene encoding a phosphatase involved in actin filament organization and endocytosis, were also found to suppress TDP-43 toxicity in yeast [78].

Interestingly, the deletion of *SET3*, a member of the histone deacetylase (HDAC) complex, was identified as a potent suppressor of TDP-43 toxicity. Subsequent studies demonstrated that HDACs can influence the aggregation and toxicity of TDP-43 by modulating the acetylation status of proteins involved in cellular stress responses in cell lines and animal models [120,121,122]. Furthermore, HDAC inhibitors, by loosening tightly packed chromatin, have been shown to increase the expression of neuroprotective genes, which enhances neuron survival in animal models of ALS. This suggests that HDAC inhibitors may represent an innovative strategy to mitigate the effects of protein aggregation in neurodegenerative diseases. For example, the role of HDACs in regulating TDP-43 pathology offers a potential therapeutic avenue, as modulating HDAC activity could influence the progression of TDP-43-related diseases [123].

## 3. Conclusions

*S. cerevisiae* has long been a model of choice for investigating fundamental cellular processes that are evolutionarily conserved, such as energy metabolism and protein function/defects. Although more complex organisms like *Drosophila melanogaster* and *Caenorhabditis elegans* have become increasingly popular, *S. cerevisiae* remains an invaluable tool in biological research due to its cost-effectiveness, time efficiency, and ease of genetic manipulation. These features make it particularly well suited for high-throughput screenings and large-scale experiments aimed at identifying potential therapeutic targets.

Many pathways involved in NDs are conserved throughout evolution, making yeast models ideal for studying neurodegeneration. For instance, *S. cerevisiae* models of TDP-43 proteinopathies have played a pivotal role in exploring ALS-related TDP-43 toxicity. These models have enabled researchers to identify enhancers and repressors (summarized in Table 1) of TDP-43-induced cell death, advancing our understanding of the disease’s mechanisms and contributing to potential therapeutic interventions.

Despite these advances, translating findings from yeast models to human therapeutic applications remains challenging due to biological differences as well as obstacles in pharmacokinetics, pharmacodynamics, and safety concerns. Overcoming these hurdles is necessary to bridge the gap between preclinical models and human therapies.

Nonetheless, *S. cerevisiae* continues to offer tremendous potential for future research. Its simplicity does not undermine its ability to model complex biological processes, as it shares critical metabolic pathways and stress response mechanisms with humans. The ease of genetic manipulation in yeast enables rapid, cost-effective experiments, and the growing use of advanced techniques like CRISPR enhances the modeling of complex diseases in *S. cerevisiae*.

Looking ahead, yeast models could be further utilized to explore gene–environment interactions in NDs and investigate new strategies for diseases like ALS, focusing on protein translation accuracy, protein folding, and the clearance of misfolded proteins. These advances, combined with evolving molecular tools, make *S. cerevisiae* an increasingly powerful model for understanding disease mechanisms and developing therapeutic approaches.

## Figures and Tables

**Figure 1 jof-11-00188-f001:**
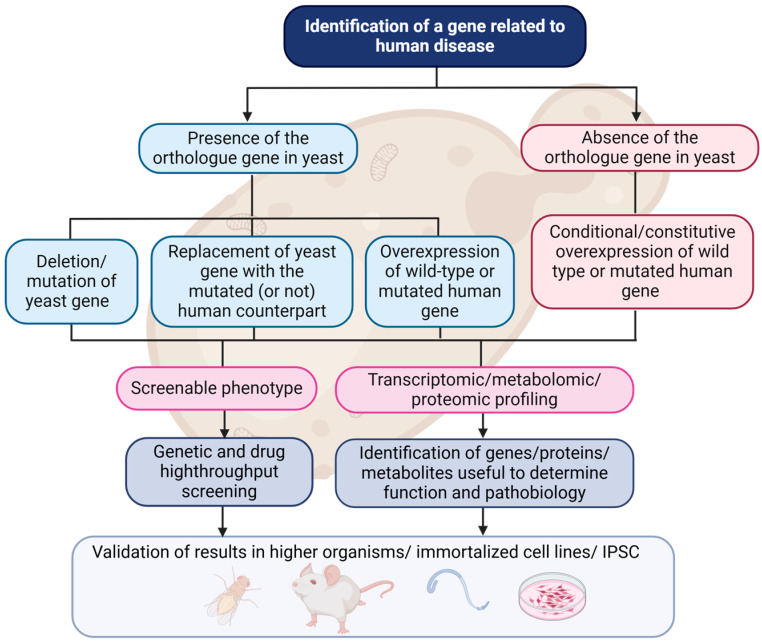
Strategy for studying human pathologies in yeast. When a mutation responsible for a human disease is identified, the first step is to determine if a homologous gene exists in yeast. If such an orthologue is found, it must be characterized. Yeast strains can be engineered by directly deleting or mutating the gene of interest, or by replacing the yeast gene with its human counterpart. Alternatively, the human protein can be overexpressed in yeast through plasmid-based methods or genome integration. In cases where no orthologue exists, the human gene product can still be introduced into yeast. Once the yeast model is generated, analysis can proceed by profiling various aspects such as growth and/or metabolic phenotypes, and large-scale transcriptome, metabolome, or proteome approaches. Yeast screens can be conducted to identify genetic modifiers or potentially druggable phenotypes. The results can then be validated in mammalian cell lines (either immortalized or induced pluripotent stem cells (IPSC)) or in higher organisms (created in BioRender. Peggion, C. (2025) https://BioRender.com/p78q779 (accessed on 10 January 2025)).

**Table 1 jof-11-00188-t001:** Yeast suppressor of yeast TDP-43 toxicity.

Gene Name	Function	Confirmed in Higher Models	Ref.
*SIS1*	Type II HSP40 co-chaperone that interacts with the HSP70 protein; shuttles between cytosol and nucleus	Yes (cortical neurons)	[124]
*STI1*	Evolutionarily conserved Hsp90 cochaperone; regulates spatial organization of amyloid-like proteins in the cytosol	Yes (N2a cells)	[87]
*HSP104*	Disaggregase; required for the protein aggregate solid-to-liquid phase transition and dispersal of liquid condensates	Yes (*C. elegans*)	[91,93,96,125,126]
*TIP41*	Negative regulator of the TORC1 signaling pathway	No	[101]
*PBP1*	Involved in P-body-dependent granule assembly; regulates TORC1 signaling and autophagy; interacts with Pab1p to regulate mRNA polyadenylation	Yes (mice, human patients, and cells)	[101,102]
*CNC1*	Cyclin-dependent protein serine/threonine kinase regulator subunit of the mediator complex; capable of sequence-specific RNA polymerase II core promoter binding during cellular response to heat	No	[111]
*DNM1*	Dynamin-related GTPase involved in mitochondrial organization	No	[111]
*PABPN1*	Human protein, poly(A)-binding protein involved in control of the length of poly(A) tails on nuclear mRNA transcripts	Yes (D. melanogaster and murine primary neurons)	[115]
*YBH3*	Protein involved in apoptosis	No	[111]
*DBR1*	RNA lariat debranching enzyme; catalyzes debranching of lariat introns formed during pre-mRNA splicing	Yes (neuronal cell lines)	[78]
*NCL*	Human protein; major nucleolar protein; found associated with intranucleolar chromatin and pre-ribosomal particles. It induces chromatin decondensation by binding to histone H1. It is thought to play a role in pre-rRNA transcription and ribosome assembly	Yes (Hek293T cells)	[83,116]
*CCE1*	Mitochondrial cruciform cutting endonuclease	None	[78]
*DOM34*	Protein that facilitates ribosomal subunit dissociation	None	[78]
*RPL16A*	RNA-binding subunit of the cytosolic large ribosomal subunit; involved in translation	None	[78]
*SET3*	Defining member of histone deacetylase complex	Yes (SH-SY5Y cells, *D. melanogaster* and mice)	[121,122]

## Data Availability

No new data were created or analyzed in this study. Data sharing is not applicable to this article.

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
