# Peer review of "A Twist in Yeast: New Perspectives for Studying TDP-43 Proteinopathies in S. cerevisiae"

_jof, 2025, doi:10.3390/jof11030188_

Round 1

Reviewer 1 Report

Since the discovery of the relationship between abnormal aggregation of TDP-43 and neurodegenerative diseases, including amyotrophic lateral sclerosis (ALS) and Alzheimer's disease, the function as well as toxicity of TDP-43 has been extensively studied. In the current manuscript, the authors nicely review the current knowledge of the neurodegenerative mechanisms associated with TDP-43 in higher eukaryotes. In addition, the article includes a set of evidence demonstrating that the budding yeast Saccharomyces cerevisiae is a valuable model to study to the cytotoxic effects of TDP-43. Each section is well-organized, and the manuscript is of high quality. 

Minor point: 

1) Figure 1 and 2(A): The font size is small, making the text difficult to read. The authors should use a larger font size.

Reviewer 2 Report

The manuscript provides and excellent overview on the relevance of Saccharomyces cerevisiae as a model to study human TDP-43 and its effects on ALS. A number of important references were not included in the work, which is understandable due to the vast literature available in the field. The key references however are present and well distributed within the different topics. 

The organisation of the work is excellent, providing a good starting point for a person entering the field. 

A few choice of words could be improved. 

Line 71 - "highly achievable"

Line 203 - "to accumulate" 

Line 235 - "Although"

Line 277 - "increased the level"

Line 369 - 'enabling to cure"

Line 380 - "to"

Line 405 - "manipulability"

Line 416 - "far away"

 Line 112: re-order transcriptome, proteome, metabolome

Line 135: (in Table 1 the list of genes definitively linked 135

to ALS are reported in orange) - confusing

Line 146: The discovery method and the number of known gene variants and people 146

affected" - people affected where and when?

Table 1 - keep either small or large caps consistent. Top alignment in the cells would make the table clearer. Correct Mg2 font

Line 171 - microRNAs (maybe plural would be best)

Line 172 - it would be clearer if the 2 processes were specified

Line 204 - re-write for clarity "a well-folded domain with one helix and six β-sheet 204

(NTD residues 1–103, Figure 2) is present"

Line 205 - specify what "It" is

Line 207 - re-organize sentence for clarity (Such...)

Figure 2 legend could be improved as colour names do not match the actual colours and several details are not spelled out in the legend 

Line 279 - last sentence is not clear

Reviewer 3 Report

Sections 1 and 2 could be combined and focused on TDP-43. Section 2 is very diffuse and introducing many items and references not directly relevant to the point of the review. TDP-43 should be the focus of section 2.

More details on the original Lindquist TDP-43 model would be useful. The description feels vague for a yeast-forward manuscript.

Table 1- while containing interesting information- distracts from the focus of the review, which is (as promised by the title) TDP-43. One can either expand the scope of the review to include yeast models for other proteinopathies beyond TDP-43, or remove this table.

Section 3.1, TDP-43’s ability to undergo LLPS should be mentioned.

Section 3.2, the TDP-43 domains and mutations that are important for toxicity should be delineated, and perhaps even included in Figure 2 or in a new table.

Section 3.3, title should read “modulators” not “modulator. More importantly, this section should also mention work with other Hsp104 homologs from other fungi, plants, protozoa, etc that expressed in cerevisiae reduce TDP-43 toxicity (For example: March , 2021 Elife).

Section 3.5, can be expanded. There is work in oxidative stress that is relevant to this section (for example: Bharathi, 2021, BBA)

Section 3.6 is mislabeled. It claims to review work involving TDP-43 modifiers beyond RNA binding proteins, when all the work presented involves RNA binding proteins. Hence, the title should be something more like “RNA binding proteins as Modifiers of TDP-43 Toxicity.”

An additional section reviewing TDP-43 toxicity modifiers beyond RNA binding proteins would be very helpful. Armakola, Nat Genetics, 2012 mentions lipid metabolism proteins, and histone deacetylase complexes. Indeed, histone deacetylase complexes’ impact on TDP-43 toxicity have been verified in mammalian systems.

A table compiling modifiers covered in Section 3, the references, and whether they have been verified in other model systems would be a useful addition.

In addition to covering the limitations of yeast models, the conclusion section should also outline the advantages of the yeast model and what further questions could be addressed by the model. Overall, the review leaves the reader wondering why yeast models should be used at all- when the intention of the review is likely the opposite.  

All throughout, carefully review yeast protein names vs. human protein names.

Minor Suggestion: Consider adding a colon (rather than a period) in the title (i.e. “A Twist in Yeast:”)

Word choice and sentence construction needs to be revised all throughout. A few examples:

Page 5, line 150: “Different are the cellular mechanisms implicated in MN cell” should be “Different cellular mechanisms have been implicated”

Page 11, line 400: “S.cerevisiae has long been an elective model” should read something like “S.cerevisiae has long been a model of choice.”

Round 2

Reviewer 3 Report

All major comments have been addressed in the revision

Table 1's format is a bit odd, perhaps adding what domain these mutations lie on (i.e. add another column for domain, and separate mutations by domain) might help.
